
# Pairing remote sensing and clustering in landscape hydrology for large-scale changes identification. Applications to the subarctic watershed of the George River (Nunavik, Canada).

Eliot Sicaud[1], Daniel Fortier[1,2], Jean-Pierre Dedieu[2,3], and Jan Franssen[4]

[1]Geography department, University of Montreal, Montreal, Quebec, Canada
[2]Centre for Northern Studies, Laval University, Quebec, Quebec, Canada
[3]Institute of Environmental Geosciences, University of Grenoble-Alpes/CNRS/IRD, 38058 Grenoble, France
[4]Geomorphix, 83 Little Bridge St. Unit 12, Almonte, Ontario, Canada

**Correspondence:** Eliot Sicaud (eliot.sicaud@umontreal.ca)

**Abstract.** For remote and vast northern watersheds, hydrological data are often sparse and incomplete. Landscape hydrology provides useful approaches for the indirect assessment of the hydrological characteristics of watersheds through analysis of landscape properties. In this study, we used unsupervised Geographic Object-Based Image Analysis (GeOBIA) paired with the Fuzzy C-Means (FCM) clustering algorithm to produce seven high-resolution territorial classifications of key remotely sensed

hydro-geomorphic metrics for the 1985-2019 time-period, each spanning five years. Our study site is the George River watershed (GRW), a 42,000 $km^2$ watershed located in Nunavik, northern Quebec (Canada). The subwatersheds within the GRW, used as the objects of the GeOBIA, were classified as a function of their hydrological similarities. Classification results for the period 2015-2019 showed that the GRW is composed of two main types of subwatersheds distributed along a latitudinal gradient, which indicates broad-scale differences in hydrological regimes and water balances across the GRW. Six classifica-

tions were computed for the period 1985-2014 to investigate past changes in hydrological regime. The seven-classification time series showed a homogenization of subwatershed types associated to increases in vegetation productivity and in water content in soil and vegetation, mostly concentrated in the northern half of the GRW, which were the major changes occurring in the land cover metrics of the GRW. An increase in vegetation productivity likely contributed to an augmentation in evapotranspiration and may be a primary driver of fundamental shifts in the GRW water balance, potentially explaining a measured decline

of about 1 % ($\sim 0.16$ $km^3y^{-1}$) in the George River's discharge since the mid-1970s. Permafrost degradation over the study period also likely affected the hydrological regime and water balance of the GRW. However, the shifts in permafrost extent and active layer thickness remain difficult to detect using remote sensing based approaches, particularly in areas of discontinuous and sporadic permafrost.

## 1   Introduction

Landscape hydrology, defined as the study of the movement and storage of water in landscapes (Ferguson, 1991), provides a way to link together numerous hydrological processes. Landscape hydrology attempts to understand watershed hydrological regimes by looking directly at the landscape features that drive them, as opposed to watershed hydrology, that traditionally





conceptualizes watersheds as event recorders, with precipitation as input and evapotranspiration (ET) and runoff as output (Gao et al., 2018). Among these landscape features, three have a predominant control on watershed hydrological regimes (Gao

et al., 2018): topography, land cover, and soil and geology (Fig. 1).

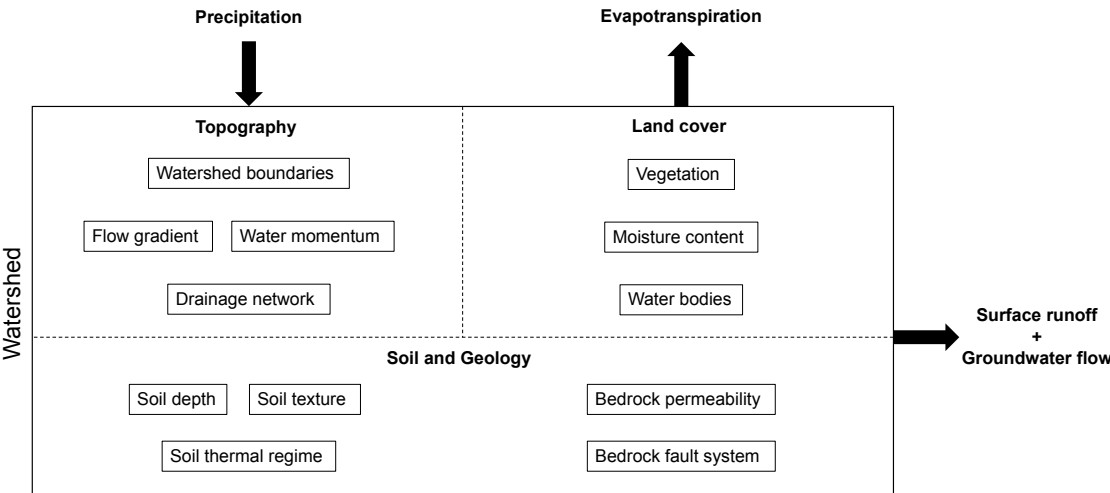

**Figure 1.** The conceptual framework used in this study provided by a landscape hydrology approach. Three main types of landscape features exert control on the hydrological regime of a watershed: topography, land cover, and soil and geology (Gao et al., 2018).

Hydrological processes such as precipitation, runoff, storage, or ET interact to determine local runoff and storage patterns and, over time, these interactions define the hydrological regimes of a watershed's spatial units, from individual hillslope source areas up to an entire watershed. The hydrological regime defines the general characteristics of the range of hydrological responses.

The general water flow is controlled by topography as the latter delineates the watersheds boundaries, sets the local microclimate, determines the flow gradient and water momentum, and settles the drainage network (Freeze and Harlan, 1969; Costa-Cabral and Burges, 1994; Young et al., 1997; Devito et al., 2005). Land cover refers to the spatial distribution of land use, such as vegetation, lakes, wetlands, or snow cover, and impacts nearly all aspects of the hydrological processes (Gao et al., 2018). The general water balance and ET are strongly influenced by vegetation, and infiltration processes are affected by

leaf intercepting precipitation, the hydraulic conductivity of the sediments, macro-pores created by root channels and cracks, and soil evolution over time (Tabacchi et al., 2000). ET and water storage are also impacted by the presence of lakes and wetlands (FitzGibbon and Dunne, 1981). In cold regions, the nival regime of watersheds is characterized by snow storage and snowmelt. Infiltration and water retention processes are controlled by soil types, depth, texture, permafrost distribution and active layer properties, and antecedent moisture conditions, whereas groundwater storage and movement are determined by

bedrock permeability and fault systems (Saxton et al., 1986; Devito et al., 2005; Hinzman et al., 2006; Viville et al., 2006).

Considering the control of all these landscape features on the hydrological regime of a watershed, key hydro-geomorphic metrics can be measured to assess the influence of landscape on hydrological regime and water balance. Because landscape





heterogeneity plays a crucial role in the spatial distribution of these different controls, the metrics must be measured at the watershed scale to consider hydrological processes in their entirety. Field measurements are typically local and rarely permit

relatable extrapolations at the watershed scale, but remote sensing can be used to obtain landscape metrics at the watershed scale for hydrological studies (Schultz, 1988; Bring et al., 2016; Gao et al., 2018). For instance, Digital Elevation Models (DEM) parametrize topography, optical and multi-spectral imagery is used to categorize land cover features, and digitized high-resolution maps, derived from field measurements and remotely sensed data, can be used as good approximations for the spatial distribution of soil and geology.

One way to deduce clear hydrological regimes from heterogeneous landscape features is to use classifications and similarity analyses (McDonnell et al., 2007; Wolfe et al., 2019; Hashemi et al., 2022), as they help describe patterns by filtering unimportant details and focusing on emergent properties, and objectively assessing resemblance between complex objects. Heterogeneity is present at different spatial scales and segmenting the studied watershed into smaller subwatersheds and classifying these units in similarity clusters provide a way to interpret patterns of hydro-geomorphic metrics at the subwatershed,

cluster of subwatersheds, and watershed scales.

In this study we present a method based on landscape hydrology to assess the hydrological regime of a watershed using a clustering approach of remotely sensed data. The study site is the George River Watershed (GRW) situated in Nunavik (Fig. 2), the subarctic region of northern Quebec (Canada). The George River drains an area of more than 42,000 $\text{km}^2$ (about the size of Switzerland), begins its course in the sporadic permafrost zone of the boreal forest, north-east of Schefferville, and discharges

into the Ungava Bay, about 600 $\text{km}$ north, in the continuous permafrost tundra environment (from 54.5° N to 59° N, and 63.5° W to 66.5° W) (Fig. A1). The GRW is localized in the hydro-physiographic region of the Canadian Shield, characterized by the presence of bedrock constituted of igneous and metamorphic rocks (Heath, 1988). The local relief is generally low, rarely exceeding 100 m, with isolated hills standing above the George Plateau and the Central Lake Plateau, and the elevation can exceptionally exceed 1000 m in the Torngat range located in the north-east of the GRW (Vincent, 1989). The contemporary

landscape is characterized by the predominance of exposed bedrock in the highlands (mostly in the northern half of the GRW), soils in the valleys (valley fill) and a large quantity of lakes, notably in the south of the GRW. Surface deposits in the GRW are principally composed of glacial deposits (till, present all over the GRW), glaciofluvial deposits (sand and gravel, mostly present in the southern half of the GRW), and fine glaciolacustrine and marine deposits (silt and clay, only present in the northern half of the GRW).

Hydrological data, such as precipitation and discharge, are sparse and incomplete in the GRW. Two hydrometric stations are present in the watershed, one based downstream (Helen Falls) which acquired data from 1962 to 1979, and one based halfway through the George River course (Lac de la Hutte Sauvage) which acquired data from 1975 to 1996 and from 2008 to present (Fig. B1, B2 and B3). Typical of remote stations in harsh environments, the time series of these stations have large data gaps which make them insufficient for this study which focuses on the entire GRW for the period 1985-2019. Considering

the scarcity of data, landscape hydrology stands out as the most adapted approach for the study of the GRW's hydrological characteristics, and in general for research on remote and vast northern watersheds.



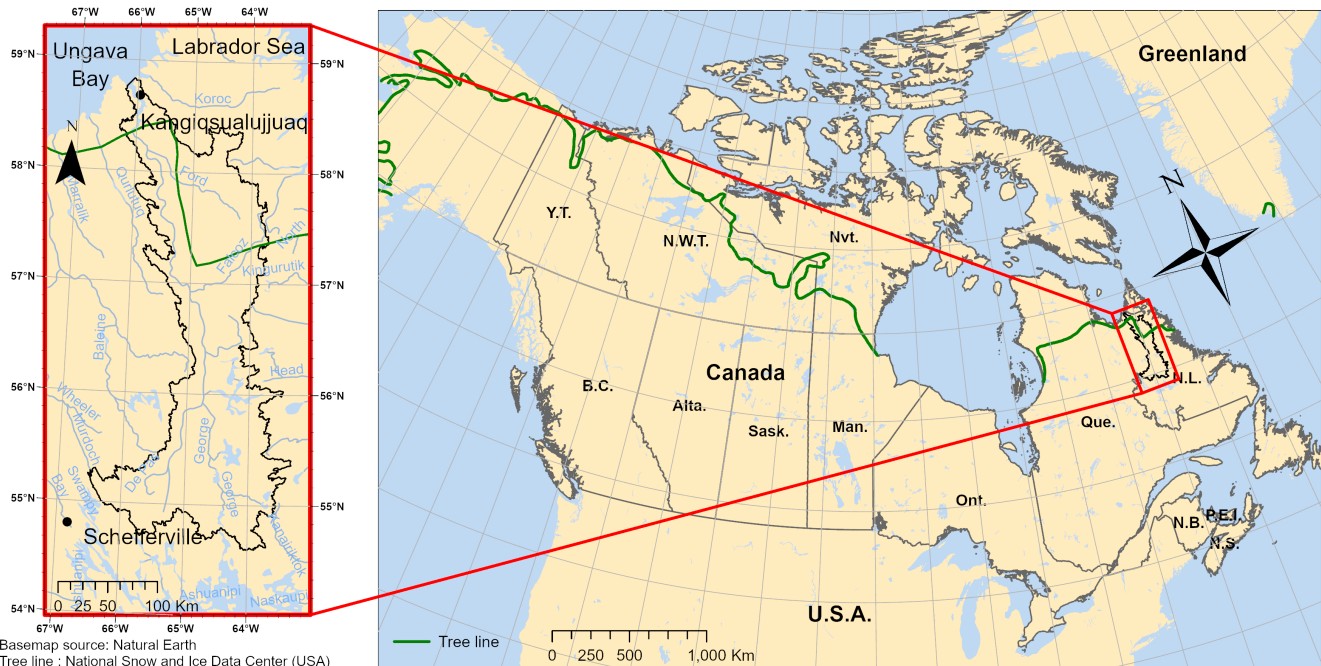

**Figure 2.** Map of the George River Watershed (GRW) situated in Nunavik, the northernmost region of Quebec (Canada). The northern tree line is indicated by a green line (National Snow and Ice Data Center, USA). The GRW covers an area of more than 42,000 $\text{km}^2$ (about the size of Switzerland).

To address the challenge of data scarcity, we used an unsupervised Geographic Object-Based Image Analysis (GeOBIA) combined with the Fuzzy C-Means (FCM) clustering algorithm to produce high-resolution territorial classifications of the GRW. Each classification spans a five-year period and is based on key remotely sensed hydro-geomorphic metrics: (i) to-
pography, (ii) land cover and (iii) geological metrics. The subwatersheds included in the GRW served as the objects of our GeOBIA and were classified in function of their hydrological similarities, which are defined as the resemblances in the hydro-geomorphic structure of the subwatersheds. A total of seven classifications were run to span the period 1985-2019. The most recent classification results provided the current hydrological configuration of the GRW, and the seven-classification time series provided understanding of the landscape changes that occurred in the last 35 years. To better grasp the nature of these changes,
a Tasseled Cap (TC) trend analysis based on Landsat images dataset was performed to assess land cover changes at a resolution of 30 m.

Using this methodology, we define the landscape properties that are likely to influence the hydrological processes at work in a large subarctic watershed to infer how properties and processes are changing and affecting the watershed hydrological regimes over time. Our hypothesis is that the hydrological regimes of the GRW are modified by changes in the watershed landscape,
themselves partly modified by climate changes. The objectives of our study were to (i) describe the current hydrological



configuration of the GRW, (ii) assess the land cover changes in the GRW over the last 35 years, and (iii) estimate the impacts of land cover changes on the hydrological regime and response, and water balance of the GRW.

## 2 Datasets and methods

We present in Table 1 the datasets used for the delineation of the GRW and its subwatersheds, the extraction of the hydro-
geomorphic metrics and the computation of the Tasseled Cap (TC) trend. In the following subsections, the complete method-
ology to produce our GeOBIAs and compute the TC trends are presented.

### 2.1 Watershed and subwatersheds delineations

The delineation of the George River watershed (GRW) was computed from the Canadian Digital Elevation Model (CDEM) dataset using the hydrology tools of the Spatial Analyst Toolbox in ArcMap 10.6.1. The GRW subwatersheds' delineations
were computed from the resulting Digital Elevation Model (DEM) clipped with the GRW delineation, using the ArcHydro Toolbox extension in ArcMap 10.6.1. The DEM was first leveled and reconditioned by burning the CanVec lakes and streams shapefiles over it. The outlet of each subwatershed was then set at every confluence where the affluents were draining an area of at least 100 $km^2$, resulting in the delineation of 218 subwatersheds contained in the GRW, with sizes ranging from 6 to 773 $km^2$ with an average of 193 $km^2$.

### 105 2.2 Computation and extraction of hydro-geomorphic metrics

A total of 10 hydro-geomorphic metrics were computed and extracted from our subwatersheds delineations. These metrics were categorized into three types: topographic, land cover and geological metrics, respectively (Table C1).

Our topographic metrics contain the mean elevation, the mean slope, the drainage density, and the form factor of our 218 subwatersheds. The computation and extraction were completed using the CDEM dataset, the Spatial Analyst Tools and the
ArcHydro Toolbox extension in ArcGIS 10.7.1, and a Python script.

For land cover metrics, a total of three spectral indices were computed : (i) the normalized difference vegetation index (NDVI), correlated to vegetation density (Tucker, 1979); (ii) the normalized difference moisture index (NDMI), correlated to soil and vegetation moisture content (Gao, 1996); and (iii) the normalized difference water index (NDWI), correlated to the presence of surface water, including lakes, streams, but also snow (McFeeters, 1996). Details on indices computation are
presented in Appendix D.

Geological metrics were derived from the surface deposit map of northern Quebec. Since groundwater conditions in the GRW are affected by the presence of permafrost (Heath, 1988), the types of surface deposits were classified as thaw-stable or thaw-sensitive deposits by L'Hérault and Allard (2018). Thaw-stable deposits are composed of exposed bedrock or bedrock overlaid by a thin layer of surficial deposits and ice-poor surficial deposits containing little ice, such as sand or gravel. Thaw-
sensitive deposits include ice-rich deposits containing a considerable proportion of fine particles (i.e., silt and clay), such



**Table 1.** Summary of the datasets used in this study.

| Datasets | Use | Resolution/Scale | Additional information |
| --- | --- | --- | --- |
| Canadian Digital Elevation Model (CDEM) | – Watershed and subwatersheds delineation<br>– Topographical metrics computation | – 0.75 arc seconds base resolution (about 20 m in plane coordinate projection) and a 0 to 10 m range of altimetric accuracy for the GRW region<br>– Reprojected at a 30 m resolution | Produced by Natural Resources Canada (https://open.canada.ca/data/en/dataset/7f245e4d-76c2-4caa-951a-45d1d2051333). |
| Lakes and streams shapefiles | Watershed and subwatersheds delineation | 1/50,000 scale | From the CanVec database, produced by Natural Resources Canada (https://open.canada.ca/data/en/dataset/9d96e8c9-22fe-4ad2-b5e8-94a6991b744b). |
| Collection of Landsat surface reflectance images | – Land cover metrics computation<br>– Tasseled Cap (TC) trend analysis | 30 m resolution | – From Landsat-5 TM, Landsat-7 ETM+ and Landsat-8 OLI Collection 2 surface reflectance images with less than 70 % of cloud cover and taken within the growing season (July to August), between 1985 and 2019. Cloudy pixels were masked using the quality assessment (QA) band.<br>– Acquired using the Google Earth Engine (GEE) Data Catalog. |
| Surface deposit map of northern Quebec | Geological metrics computation | Every surface deposit unit larger than 100 ha | – Produced by the Ministère des Forêts, de la Faune et des Parcs du Québec (https://www.donneesquebec.ca/recherche/dataset/carte-des-depots-de-surface-du-nord-quebecois/resource/db364178-0d70-47db-83a8-0e912d7ec65).<br>– Based on geomorphologic interpretations of RapidEye images (6.5 m resolution) taken between 2010 and 2013. |
| Reanalysis climate data | Climate data comparison with land cover metrics time series | Watershed scale | – Produced by Environment and Climate Change Canada (https://climatedata.ca/).<br>– Derived from modeled historical data (1950-2005) and climate projection models (2006-2020).<br>– Only the Mean Annual Temperature (MAT) and the Total Annual Precipitation (TAP) variables were acquired.<br>– Each dataset is divided into four subwatersheds covering the entire George River Watershed, one for the Lower George region (north) and three for the Upper George region (south). |





as subglacial till, glaciolacustrine, marine, and organic deposits, which allow for the formation of segregation ice. For each subwatershed, the fractional coverages of thaw-stable and thaw-sensitive deposits were computed using a Python script.

## 2.3 Fuzzy c-means clustering

The unsupervised GeOBIA is based on the methodology of Choubin et al. (2017). The ten hydro-geomorphic metrics were
used as the input variables of the GeOBIA, and the subwatersheds were used as the objects to classify. After computation and extraction, the input variables were normalized to assure an equivalently weighted classification. The clustering algorithm used was the Fuzzy C-Means (FCM) algorithm (Dunn, 1973; Bezdek, 1981), which provides, for each object, a set of membership coefficients corresponding respectively to each cluster. Each classification returned a Fuzzy Partition Coefficient (FPC), which described how well-partitioned our dataset was. More details on the Fuzzy C-Means algorithm are presented in Appendix E.
The complete clustering methodology is summarized in Fig. 3.

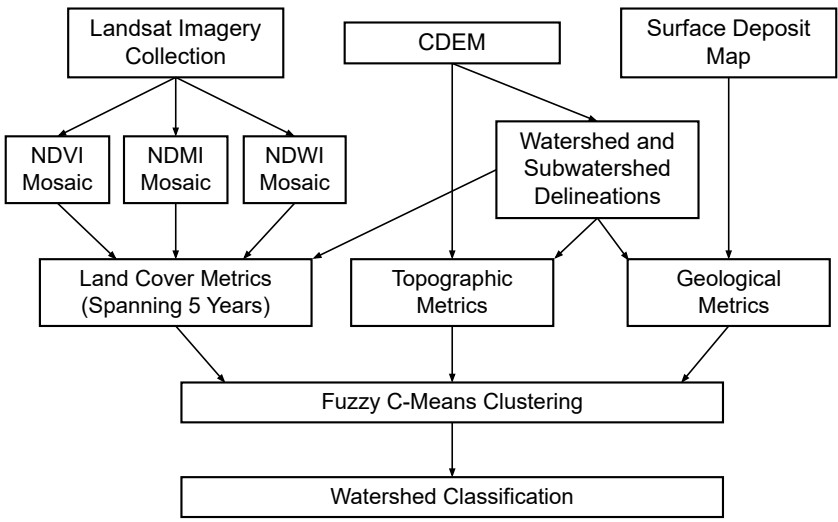

**Figure 3.** Simplified unsupervised Geographic Object-Based Image Analysis (GeOBIA) flow chart method.

## 2.4 Land cover metrics and membership coefficient trend computation

At the watershed scale, while land cover metrics can change within an interval of 35 years, topographic and geological metrics can be expected to remain relatively unchanged. Permafrost conditions maybe expected to change at the decadal-to-centennial time scale, however no products are available to quantify change in its distribution. Thus, for each subwatershed, trends in the
land cover metrics were computed for the 1985-2019 time-period. Our seven-classification time series, with each classification differing only by its input land cover variables, was used to investigate changes observed in the clustering results. To high-





light principal differences between these seven classifications, positive trends in membership coefficients were computed and selected for each cluster, with the same method used for the land cover metrics.

## 2.5 Tasseled cap trend analysis

To better understand the nature of land cover changes within the subwatersheds, a TC trend analysis was performed using our Landsat collection and following the Landsat Arctic Rgb CHanges (LARCH) method introduced by Fraser et al. (2014). This method was appropriate for the identification of large-scale changes, while keeping a 30 m resolution. Details on the TC trend computation are presented in Appendix F.

## 3 Results

The results are divided into three parts. First, classification results with the most recent data (2015-2019) are presented to describe subwatershed-scale patterns in hydro-geomorphic metrics across the GRW. Second, six other classifications were run for the period 1985-2014 and results were combined with the 2015-2019 classification to identify trends in land cover metrics and membership coefficients and assess land cover changes that occurred over the last 35 years. Third, TC trends were analyzed to characterize fine scale (resolution of 30 m) patterns in land cover changes within the subwatersheds.

### 3.1 Clustering results (2015-2019): partitioning the George River watershed

Clustering results for the period 2015-2019 showed an optimal number of two clusters of subwatersheds, Cluster 1 (C1) and Cluster 2 (C2), with a Fuzzy Partition Coefficient (FPC) of 0.6299, implying that the GRW was composed of two distinct types of hydrologically similar subwatersheds. The relation between FPC and the number of clusters identified (from 2 to 20) revealed a rapid decrease in FPC as the number of clusters increased (Fig. 4a). The clustering distribution within the GRW

displayed C1 dominating the southern part and C2 dominating the northern part of the GRW, respectively (Fig. 4b, 4c and 4d). Membership coefficients for C1 and C2 ranged from 0.072 to 0.95 and from 0.053 to 0.93, respectively, which involved a slightly greater membership for subwatersheds included in C1.

Boxplots comparing the hydro-geomorphic metrics statistical distribution by cluster (Fig. 5a, 5b and 5c) provided insight into which metrics were driving the subwatershed clustering patterns. For instance, mean elevation medians were 483 m and

527 m for C1 and C2, respectively (Fig. 5a). Mean elevation in C1 had a $\sim 200$ m range, compared to C2 with a $\sim 600$ m range, if we exclude outliers. The mean slope showed median values of 5.27 ° and 9.55 ° for C1 and C2, respectively (Fig. 5a). Drainage density statistical distributions were contained in values of $\sim 2$ $km^{-1}$ in both clusters due to the existence of high-value outliers (reaching 11.8 $km^{-1}$ in C1) (Fig. 5a). Form factor statistical distributions were similar in both clusters, meaning the distribution of the form of the subwatersheds was homogeneous in the GRW (Fig. 5a). Vegetation-related metrics, such as

mean NDVI and NDVI fractional coverage presented 0.615 and 90.8 % medians for C1, and 0.516 and 92.5 % medians for C2, respectively (Fig. 5b). For water-related metrics, mean NDMI displayed medians of 0.162 for C1, and -0.041 for C2 (Fig. 5b). NDWI fractional coverage medians were similar in both clusters, with 10.5 % for C1 and 9.2 % for C2, but C1 spanned

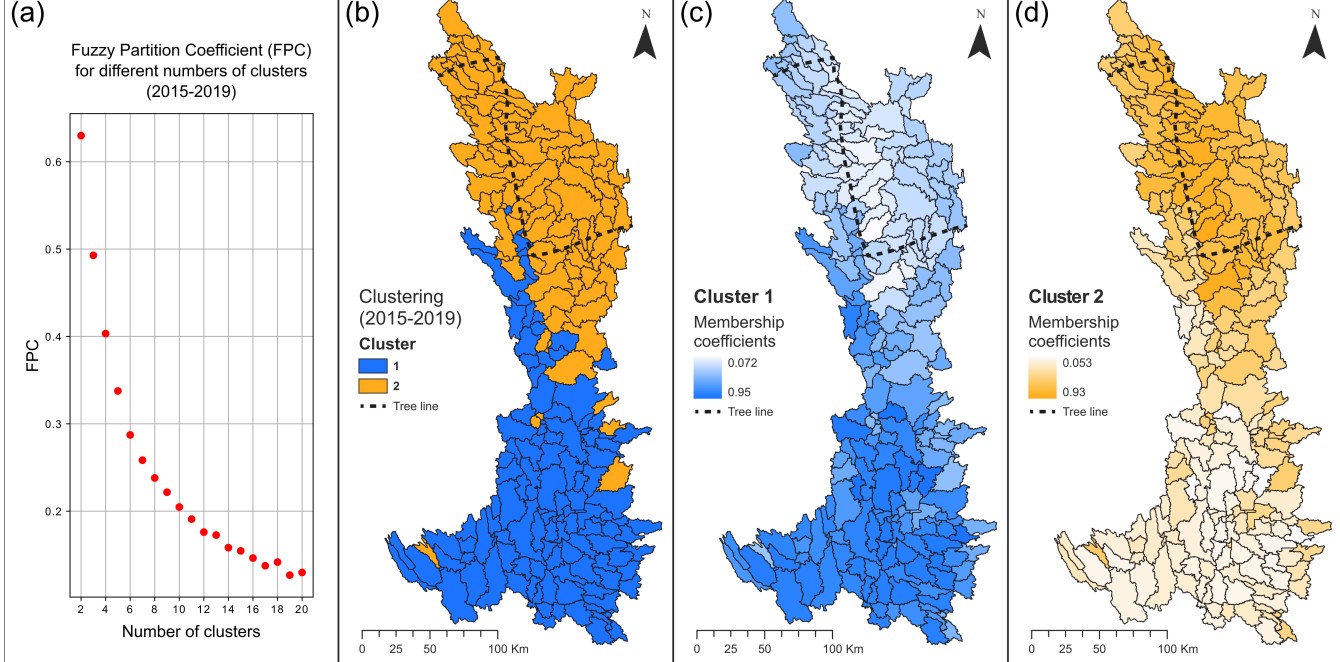

**Figure 4.** (a) Fuzzy Partition Coefficient (FPC) in function of number of clusters for the 2015-2019 classification. The highest FPC is reached for a two-cluster classification. The two clusters of subwatersheds are distributed latitudinally in the George River Watershed: (b) "hard" clustering results, (c) fuzzy clustering results for Cluster 1 and (d) fuzzy clustering results for Cluster 2. The northern tree line is indicated by a black dotted line.

a larger range reaching $\sim 35$ % when excluding outliers (Fig. 5b). Geological metrics showed a 12.0 % thaw-stable surface deposits cover median and a 71.9 % thaw-sensitive surface deposits cover median for C1, compared to a 35.5 % thaw-stable surface deposits cover median and a 54.4 % thaw-sensitive surface deposits cover median for C2 (Fig. 5c).

### 3.2 Trends in land cover metrics and clustering results (1985-2019)

Our analysis detected widespread upward trends in vegetation productivity across the entire GRW as shown by increases in mean NDVI across all subwatersheds (Fig. 6a). Trends in mean NDVI had rates ranging from 0.00052 $y^{-1}$ to 0.0016 $y^{-1}$. The highest trends were condensed in the north-west (George River's estuary), while the lowest ones were present in the north-eastern (subarctic alpine tundra) and southernmost (boreal forest) parts of the GRW. A different distribution pattern was observed for trends in the NDVI fractional coverage (Fig. 6b), but positive trends were also present in all subwatersheds. For NDVI fractional coverage, rates ranged from 0.0030 %$y^{-1}$ to 0.21 %$y^{-1}$. The highest slope values were concentrated in the north-east (Torngat Mountains foothills) and south-east (eastern George Plateau) and the lowest ones were present in the southernmost part of the GRW (George Plateau and Central Lake Plateau).



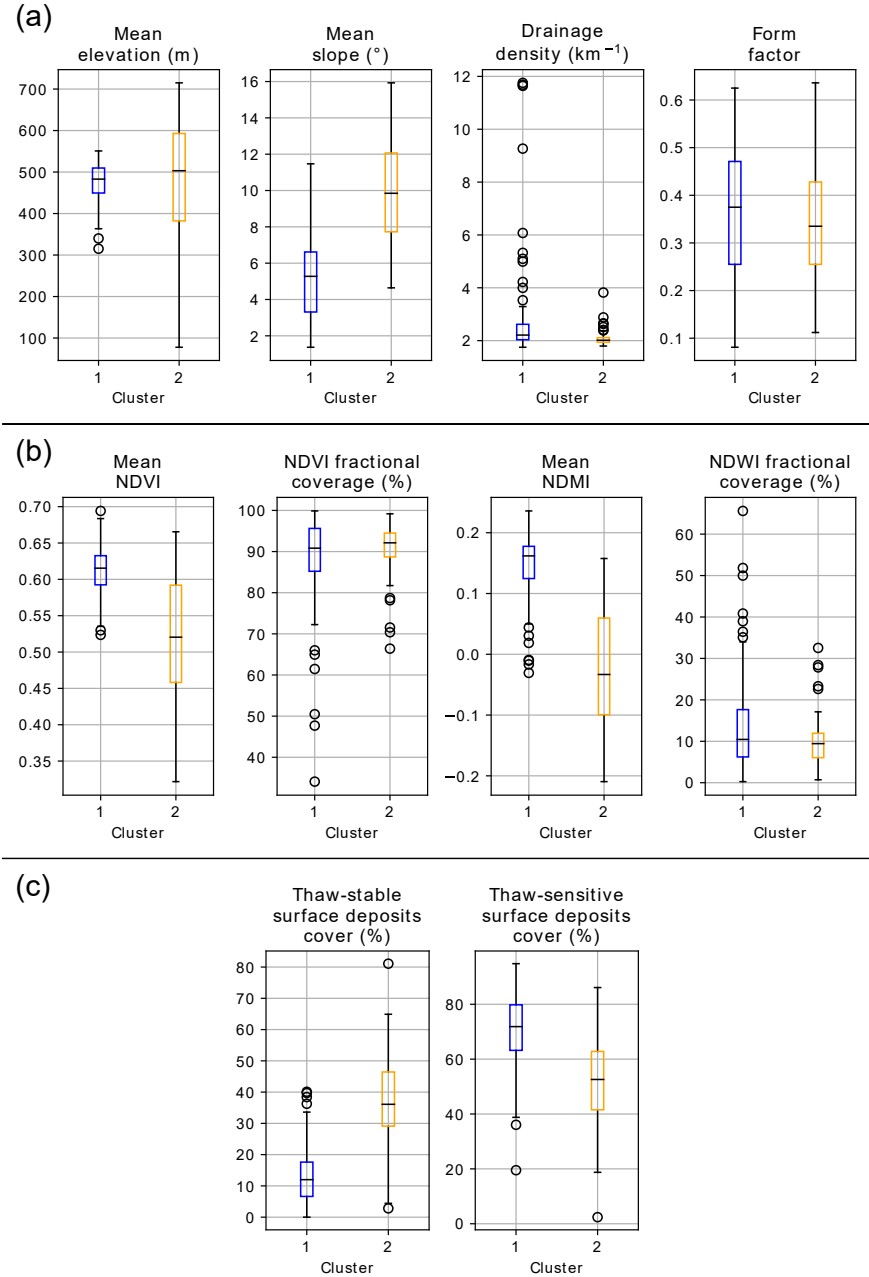

**Figure 5.** George River Watershed statistical distribution of hydro-geomorphic metrics by cluster (Cluster 1 in blue and Cluster 2 in orange) for the period 2015-2019. (a) Topographic metrics: mean elevation , mean slope, drainage density and form factor. (b) Land cover metrics: mean NDVI, NDVI fractional coverage, mean NDMI and NDWI fractional coverage. (c) Geological metrics: thaw-stable surface deposits cover and thaw-sensitive surface deposits cover.





Increases in soil and vegetation moisture content were also observed in all the subwatersheds of the GRW, as shown by increases in mean NDMI (Fig. 6c) which form a pattern similar to the mean NDVI trends. Trends ranged from 0.00023 $y^{-1}$ to 0.0014 $y^{-1}$. The highest trends were in the north-west, along the George River's main stem and near its estuary, and the lowest ones were found in the subarctic alpine tundra in the north-east (Torngat Mountains foothills) and the boreal forest in the south (George Plateau).

There were widespread downward trends in the extent of snow or water bodies across the GRW subwatersheds as indicated by trends in NDWI fractional coverage (Fig. 6d). Only one small subwatershed (6.64 $\mathrm{km}^2$), located in the south-western George Plateau, present an increase of 0.0018 $\%y^{-1}$ in NDWI fractional coverage. Negative trends in the NDWI fractional coverage ranged from -0.047 $\%y^{-1}$ to -0.00089 $\%y^{-1}$, indicating a small decrease in the extent of water bodies or snow during the summer season. The highest trends in magnitude were located in the north-east (Torngat Mountains foothills) and south-east (eastern George Plateau).

A total of 110 subwatersheds showed positive trends in their membership coefficient for C1, with a range of 0.000043 $y^{-1}$ to 0.0026 $y^{-1}$ (Fig. 6e). They were mainly gathered in the northern half of the GRW (approximate location of C2 for the period 2014-2019). Comparatively, 108 subwatersheds displayed a positive trend, with a range of 0.000029 $y^{-1}$ to 0.0027 $y^{-1}$ in their membership coefficient for C2 (Fig. 6e) and were situated in the southern half of the GRW (approximate location of C1 for the period 2014-2019).

### 3.3 Tasseled Cap trend results: detecting changes in the GRW over 35 years

TC trend analysis of Landsat satellite images (1985-2019) identified fine scale spatial patterns in landscape change within each of the subwatersheds and these patterns were consistent with the results of our clustering analyses. Using the color scheme proposed by Fraser et al. (2014) for detecting landscape changes in northern environments (Fig. G1), we found three distinct types of landscape changes as revealed on the TC trend RGB color composite images (Fig. 7).

The most obvious change was an increase in vegetation productivity characterized by increases in TC Greenness and Wetness and a decrease in TC Brightness, which can be associated with an augmentation of leaf biomass (Fraser et al., 2014). This greening trend, known as *tundra greening*, appeared in teal color in the TC trend image and was essentially present in the northernmost part of the GRW, surrounding the estuary, the George River's major stem and its tributaries (Fig. 7a).

Results also presented areas witnessing decreases in TC Brightness and Wetness and an increase in TC Greenness, which appeared in green in the TC trend image. This is generally linked to receding snow and ice (Fraser et al., 2014). This modification in land cover was more present in the north-eastern mountainous part of the GRW, but it could also be observed, to a lesser extent, in upland areas located all over the northern half of the GRW (Fig. 7b).

Another major change that occurred in the GRW was the increase in TC Wetness and decreases in TC Brightness and Greenness, which can be related to an increase of water content in soil and vegetation. This change appeared in dark blue on the TC trend image and was mostly observable around the George River's main stem, in the northern half of the GRW, between areas of greening and the streams (Fig. 7c).



**Figure 6.** Trends in (a) mean NDVI, (b) NDVI fractional coverage, (c) mean NDMI, (d) NDWI fractional coverage and (e) membership coefficients (110 subwatersheds with a positive trend in their membership to Cluster 1, compared to 108 subwatersheds for Cluster 2), for the period 1985-2019. A darker color represents a higher trend in magnitude. The northern tree line is indicated by a black dotted line.

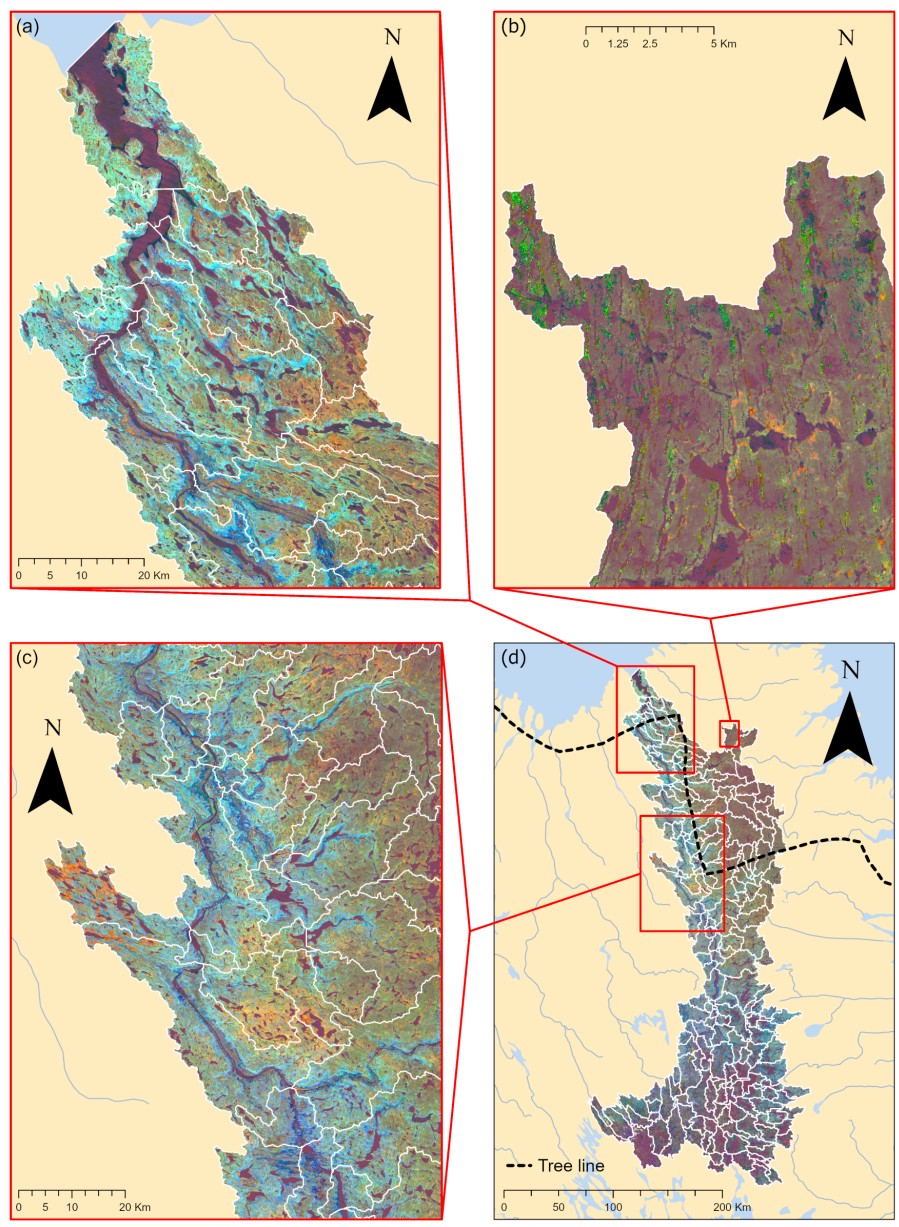

**Figure 7.** Tasseled Cap (TC) trend image of the lower course and estuary of the George River during the growing season for the period 1985-2019, produced following the LARCH method introduced by Fraser et al. (2014). (a) Areas in teal experienced an augmentation in vegetation productivity; (b) areas in green experienced a decline in albedo, moisture content in soils and vegetation, and an increase in vegetation productivity; and (c) areas in dark blue experienced an increase in moisture content in soil and vegetation. (d) Complete overview of the TC trend image applied to the GRW. The northern tree line is indicated by a black dotted line.





## 4    Discussion

In this section, we first present the current hydrological configuration of the GRW based on our landscape hydrology ap-
proach. Second, we discuss the observed land cover changes that occurred in the GRW over the last 35 years using our
seven-classification time series and our TC trend analysis. Third, we evaluate the potential impacts of these land cover changes
on the hydrological regimes of the GRW.

### 4.1    Current hydrological configuration of the GRW

With the clustering results of our GeOBIA for the period 2015-2019 (Fig. 4b, 4c and 4d), it is now possible to assess the typical
hydrological configurations of both clusters, and of the overall GRW. The hydro-geomorphic metrics statistical distributions
previously presented (Fig. 5) focus on metrics with high partitioning effects on the dataset which give insights about the general
regime, water balance and hydrological response characterizing both Cluster 1 (C1) and Cluster 2 (C2) and allow an efficient
comparison of these clusters.

Snowmelt is the major hydrological event affecting the runoff regime in the GRW (FitzGibbon and Dunne, 1981). Thus, the
GRW's annual hydrograph presents high discharge in early summer, followed by subdued hydrological responses to rainfall
events that interrupt the summer baseflow; and a runoff decrease in winter induced by an increase in snow storage.

In general, C1 includes the southernmost subwatersheds of the GRW and covers the subregions of the George Plateau and
parts of the Central Lake Plateau. The landscape of these subwatersheds is characterized by low relief variability with an
average elevation of $\sim 450$ m. Vegetation is widespread and dense ($\sim 90$ % vegetation cover and mean NDVI of $\sim 0.6$),
with the presence of a boreal forest in the south, and fine thaw-sensitive deposits are dominant in the lowlands ($\sim 70$ % fine
thaw-sensitive deposits cover compared to $\sim 15$ % thaw-stable deposits cover). The low variability of topography confers to C1
complex systems of extensive interconnected lakes ($\sim 15$ % water bodies cover) and wetlands. Ground thermal conditions range
from discontinuous permafrost in the north of C1 to sporadic permafrost in the southern lowlands, but discontinuous permafrost
is largely dominant in C1 (Fig. A1). Hydrological conditions (infiltration, runoff) are therefore less affected by permafrost than
in C2. Nevertheless, fine-grained, ice-rich permafrost in the lowlands plays a role in the distribution of wetlands in C1.

In terms of the measured hydro-geomorphic metrics in C1, the low range in mean elevation and the lower values in mean
slope (Fig. 5a) underline both the low inter-watershed and intra-watershed variability of topography. Mean NDVI and mean
NDMI (Fig. 5b) show higher values in C1, compared to C2, and relate to the presence of more abundant vegetation and more
water content in soil and vegetation, as C1 is located at lower latitudes. Where permafrost is present, thaw-stable surface
deposits cover is considerably lower (Fig. 5c) because of exposed bedrock and bedrock overlaid by a thin layer of deposits
are less present in C1. On the contrary, thaw-sensitive surface deposits cover is higher (Fig. 5c) because of the predominance
of fine deposits in the lowlands of C1. The larger number of outliers for the drainage density (Fig. 5a) and NDWI fractional
coverage (Fig. 5b) is mainly due to the presence of large lakes, which can generate high values in both metrics.

As C1 predominates in the southern part of the GRW, it experiences higher Mean Annual Temperature (MAT) (Fig. 8),
indicating late snow season and earlier spring snowmelt compared to C2. In addition to snow storage, the presence of extensive



lakes contributes considerably to storage in the general water balance of C1 (FitzGibbon and Dunne, 1981; Spence, 2000). Infiltration and retention processes are more likely to happen in C1 subwatersheds, which present high water content in soil and vegetation, more abundant vegetation (Dunne et al., 1991; Tabacchi et al., 2000; Thompson et al., 2010; Beven and Germann, 2013; Gao et al., 2018) and fine surface deposits within discontinuous to sporadic permafrost (L'Hérault and Allard, 2018).

In addition, higher MAT and the presence of high water content in soil and vegetation, dense vegetation and large lakes in C1 likely increase the contribution of ET to the water balance, as opposed to C2 (Bosch and Hewlett, 1982; Kelliher et al., 1993; Savenije, 2004; Rouse et al., 2008; Yang et al., 2009; Gerrits et al., 2010). The dominance of gentle slopes, complex interconnected lakes systems and the abundance of infiltration and retention processes indicate that input water likely has a relatively longer travel time and runoff generation is primarily controlled by pre-existing storage conditions and fill-and-spill

drainage processes (Spence, 2000; Devito et al., 2005).

C2 includes the northernmost and both the least and most elevated subwatersheds of the GRW, due to proximity to the estuary in the north-west and the Torngat Mountains in the north-east. In general, vegetation in C2 is also widespread but less dense ($\sim 90$ % vegetation cover and mean NDVI of $\sim 0.5$) and C2 has more thaw-stable deposits and less fine thaw-sensitive deposits than C1 ($\sim 35$ % thaw-stable deposits cover and $\sim 50$ % fine thaw-sensitive deposits cover in total in C2). Landscapes

of the subwatersheds in C2 can be characterized by two distinct types of physiographic units: the uplands and the soil-filled valleys. In the uplands, with elevation reaching 981 m, vegetation is sparse, and bedrock is mostly exposed or overlaid by a thin layer of surficial deposits. Deposits are thus categorized as thaw-stable, and the ground thermal regime is controlled by the presence of continuous permafrost (Fig. A1). In the soil-filled valleys, incised by the George River's main stem and its major tributaries, vegetation and fine-grained thaw-sensitive deposits are more abundant, and the distribution of permafrost is

discontinuous (Fig. A1).

In terms of the measured hydro-geomorphic metrics in C2, the extensive range in mean elevation and the higher values in mean slope (Fig. 5a) highlight the important inter-subwatershed and intra-subwatershed relief variability, respectively. Lower values in mean NDVI and mean NDMI (Fig. 5b) attest the lower abundance of vegetation in C2, which is due to its northern location and the predominance of a colder climate at high elevations. Because the uplands present a large portion of exposed

bedrock and bedrock overlaid by a thin layer of surficial deposits, thaw-stable surface deposits cover is higher in C2 than C1 (Fig. 5c). However, thaw-sensitive surface deposits cover is larger than thaw-stable deposits cover in C2 (Fig. 5c), as subglacial till is the dominant type of surface deposits in the entire GRW.

Because C2 is situated north and its north-eastern part reaches high elevations, MAT is lower (Fig. 8) and therefore the snow season takes place earlier in fall and snowmelt occurs later during summer. In the uplands, the presence of continuous

permafrost, exposed bedrock, thin sediment layers, and sparse vegetation favors lateral flow and runoff. Infiltration is minimal and runoff is likely at snowmelt when the active layer is still frozen (no infiltration due to ice in the soil porosity). The hillslope storage capacity is low in the spring and progressively increases during the summer as the active layer thaws and subsurface lateral flow occurs (Kane et al., 1991; Hinzman et al., 2006; Spence and Woo, 2006; Thompson et al., 2010; Chiasson-Poirier et al., 2020). In the soil-filled valleys with fine-grained sediment in discontinuous permafrost, infiltration is more frequent

and contributes to the saturation of the active layer where permafrost is present. ET has a reduced importance in the general





**Figure 8.** Reanalysis climate data (MAT and TAP) plotted against statistical distribution time series of land cover metrics (mean NDVI and mean NDMI) for the period 1985-2019.

water balance of C2, due to lower MAT and Total Annual Precipitation (TAP), sparse vegetation and less moisture in soil and vegetation (Rouse et al., 2008). Generally, the presence of steep slopes and the likeliness of runoff process where ground conditions are met, confer to C2 a relatively short travel time to input water (Devito et al., 2005), when compared to C1.

In summary, our landscape hydrology approach suggests that C1 includes headwater subwatersheds with regime punctuated by fill-and-spill processes, where water has generally longer residence times; and C2 is composed of subwatersheds located relatively near the estuary with yield including more frequent surface runoff processes and lateral subsurface flow, where water





has generally shorter residence times. These characteristics combined to the longitudinal form of the GRW generally yield smooth responses to snowmelt and precipitation events at the estuary.

## 4.2 Land cover changes over the last 35 years

The GRW experienced approximately the same climate trends over the last 35 years, with the southern part having constantly higher MAT (+1 °C) and TAP (+100 mm) values than the northern part (Fig. 8). Between 1985 and 2020, MAT increased from -5 °C to over -4 °C in the southern part, and from -7 °C to about -6 °C in the northern part; and TAP increased from 700 mm to 800 mm for the southern part, and from 650 mm to 700 mm in the northern part (Fig. 8). Overall, MAT increases showed clear trends as opposed to TAP, which experienced generally more variations than MAT. MAT acts as an important control

factor on the permafrost thermal regime, vegetation productivity and increase in soil and vegetation water content, without underestimating the contribution of TAP.

Our land cover metric trend analysis shows that the most important increases in vegetation abundance (Fig. 6a) and in soil and vegetation moisture content (Fig. 6c) happened in subwatersheds situated along the GRW main stem, near the river's mouth, which present a valley-type physiography with relatively low elevations, more abundant vegetation and fine thaw-sensitive

surface deposits, high water content in soil and vegetation, and discontinuous permafrost. On the other hand, increases in vegetated landscape cover and decreases in water bodies or snow coverage were found in nearly all the GRW (Fig. 6b and 6d), but more substantial trends in magnitude only appeared in C2's most elevated subwatersheds. In general, the northern half of the GRW experienced the highest trends in magnitude for all land cover metrics.

Positive trends in membership to C1 concerned only subwatersheds located in the northern half of the GRW (Fig. 6e), and

the highest trends were present in the subwatersheds with the lowest membership to C1 in the 2014-2019 GeOBIA (Fig. 4c). Similarly, positive trends in membership to C2 concerned only subwatersheds located in the southern half of the GRW (Fig. 6e), and the highest trends were present in the subwatersheds with the lowest membership to C2 in the 2014-2019 GeOBIA (Fig. 4d). Variations in membership coefficients arise from the displacements of the centroids of the subwatersheds and the clusters, which are induced by changes in land cover metrics. For both clusters, a similar number of subwatersheds showed increases

in their membership coefficients (110 subwatersheds for C1 and 108 subwatershed for C2), the highest trends concerned the subwatersheds with the lowest membership coefficients and the lowest trends concerned the subwatersheds with the highest membership coefficients. This can be interpreted as a homogenization of the land cover metrics across the GRW, suggesting that the northern type subwatersheds are becoming more hydrologically like the southern type, with potential augmentations of infiltration and ET, that tend to be modified by increasing vegetation productivity, an augmentation of moisture content in

soil and vegetation, and discontinuous permafrost degradation induced by higher MAT and TAP. The increase in active layer thickness and the diminution of permafrost extent favor subsurface water flow which in turn creates a positive feedback effect to permafrost thawing by increasing heat advection related to subsurface water flow (Chen et al., 2019; Chen et al., 2021).

Our TC trend analysis with a 30 m resolution sheds light on the nature of the land cover metrics trends presented above. The highest increases in vegetation related metrics happened between the periods 1995-1999 and 2000-2004, and between

the periods 2005-2009 and 2010-2014 (Fig. 8), which is in accordance with the results of Bayle et al. (2022), who found two



significant and distinct waves of greening, centered around 1996 and 2011, in the GRW. Tundra greening is principally located in the northernmost discontinuous permafrost part of the GRW, north of the tree line (Fig. 7a), where vegetation was previously sparse and low. In this region, the greening is principally caused by shrub expansion (Tremblay et al., 2012). Higher trends in mean NDVI situated in the northernmost subwatersheds above the tree line (Fig. 6a) suggest that shrub expansion is a major

contributor to the total increase in vegetation productivity in the entire GRW.

Increase in soil and vegetation moisture is spatially correlated to an increase in vegetation productivity, based on the mean NDMI trends analysis (Fig. 6a and 6c). Between 1999 and 2015, mean NDMI showed a constant increase in accordance with MAT, TAP and mean NDVI. The TC trend analysis confirms this statement since increases in TC Wetness index only appears in teal and blue colors, related to vegetation increase, and in pink color, which is not observed on the TC trend image (Fig.

7c). No causal relations can be suggested since soil and vegetation moisture content and vegetation productivity are related, influence each other in complex ways (Rodriguez-Iturbe, 2000), and also depend on ground thermal conditions which have undoubtedly changed in response to MAT and TAP increases over the study period.

The TC trend image also provides a potential explanation to the decreasing trends in NDWI fractional coverage observed in the northern half of the GRW (Fig. 6d). This decreasing trend in NDWI fractional coverage is most likely related to a decrease

in perennial-to-long-lasting snow patches. The NDWI highlights water-or-ice-covered pixels, so it cannot distinguish between whether water bodies or snow storage decreased in extent. In the TC trend image (Fig. 7b), however, we only observe green areas linked to snow or ice receding coverage, and no yellow area generally associated with drained lakes (Fraser et al., 2014). Moreover, as these green areas are contained in shaded depressions located in the uplands, where continuous permafrost and ice-poor thaw-stable deposits are dominant, permafrost thaw cannot engender the formation or the drainage of water bodies.

Accordingly, subwatersheds experiencing the most important decreasing trends in NDWI fractional coverage are in the most elevated part of the GRW, where perennial-to-long-lasting snow patches receding is most susceptible to occur.

### 4.3 Impacts of land cover changes on hydrological processes

Over the last 35 years, we observed increases in vegetation productivity and in soil and vegetation water content in the northernmost part of the GRW, and a decrease in perennial-to-long-lasting snow patches in the north-eastern uplands. These wide-spread

and potentially permanent shifts in land cover characteristics can be expected to impact hydrological processes and flow path, and thus, the hydrological regime and water balance of the GRW.

With higher MAT, the GRW is inevitably witnessing an earlier snowmelt in spring, resulting in earlier peak flows (Goudie, 2006; White et al., 2007; Bring et al., 2016). As our land cover metrics and TC trend analysis show, permanent snow stored all year long in depressions tends to disappear, modifying the water balance by reducing summer storage, particularly in the

mountainous regions.

Also, with increases in MAT, vegetation productivity and soil and vegetation moisture, the GRW is most likely experiencing an increase in ET (Nicholls and Carey, 2021), affecting the water balance by returning water directly into the atmosphere and reducing the runoff response. Recent discharge analyses at the Lac de la Hutte Sauvage hydrometric station, situated halfway through the George River course, show a small decrease of about 1 %, but non-negligible in terms of volume ($\sim 0.16 \, \mathrm{km^3 y^{-1}}$),





in mean annual discharge between mid-1970s and 2017 (Gérin-Lajoie et al., 2018). This decreasing trend was associated with reduced mean summer flows while the winter base-flow remained the same. Substantial declines in annual discharge for the rivers draining the James Bay system, which includes Ungava Bay, have already been observed in the past years, averaging a decrease of 2.5 $\mathrm{km}^3\mathrm{y}^{-2}$ in total (Déry and Wood, 2005; White et al., 2007). Our interpretation is that: (i) an augmentation of vegetation productivity during the growing season is partly responsible for this decline in summer flows and annual discharge; and (ii) permafrost degradation (active layer deepening or reduction of discontinuous permafrost cover) in response to rise of MAT over the study period favored infiltration and increased the storage capacity of the GRW (Rouse et al., 1997). Although permafrost ranges from discontinuous to sporadic in the northern soil-filled valleys and in the entire southern half of the GRW, these terrains are dominated by thaw-sensitive deposits, and, infiltration processes encouraged by permafrost thaw could result in the loss of connectivity between lakes, wetlands and streams, modifying the hydrological regime and response, and the water balance (Bring et al., 2016). These mentioned changes in the water balance of subarctic watersheds have also been observed in the alpine tundra of the Taiga Cordillera ecozone in northwestern Canada (Kershaw et al., 2022).

## 5  Conclusions

Many vast northern watersheds have been poorly studied and knowledge about their structure is insufficient to be able to understand and predict their hydrological regimes. Our landscape hydrology approach can be used to address the fundamental problem of understanding and characterizing landscape heterogeneity at broad spatial scales (McDonnell et al., 2007).

The presented method, limited to a 30 m spatial resolution, a five-year temporal resolution and the existing open source remote sensing data, has provided (i) useful insights on the key hydro-geomorphic metrics that are likely driving flow regime, hydrological response and water balance changes of a vast and remote northern watershed (42,000 $\mathrm{km}^2$, about the size of Switzerland); (ii) an assessment of general changes in land cover features that experience susceptibility to climate change; and (iii) a way to estimate the impacts of land cover changes on the hydrological regimes of watersheds.

Applied to the GRW, our study confirms our initial hypothesis by showing that the GRW is composed of two distinct types of subwatersheds distributed along a latitudinal gradient, with landscape characteristics that are likely to result in distinct subwatershed flow regimes, hydrological responses, and water balances. These hydrological regimes are susceptible to change as the landscape alters and changes in response to both climate change and the normal evolution of geosystems. The southern type of subwatersheds (C1) have structural and landcover characteristics that suggest hydrological regimes governed by fill-and-spill runoff processes, water balances with emphasized ET and water storage components, and more subdued hydrological responses. The northern type of subwatersheds (C2) have structural and landcover characteristics that suggest regimes governed by early snow season, late snowmelt and uniform runoff yield, water balances characterized by more frequent surface runoff, and stronger hydrological responses. Our seven-classification time series presents a homogenization of subwatershed types associated with an increase of vegetation productivity, commonly known as tundra greening and an augmentation of water content in soil and vegetation, all concentrated in the northern half of the GRW. Moreover, our Tasseled Cap (TC) trend analysis indicated that perennial-to-long-lasting snow patches in the northern part of the GRW showed a significant decline





or disappeared over the study period. MAT increased by about 1 °C in the entire GRW over the study period, and TAP increased about 100 mm in the southern half, and about 50 mm in the northern half of the GRW. Our results suggest that

the principal driver of these land cover changes in the GRW is temperature rise, and that as a result the hydrology of the GRW is now characterized by an earlier snowmelt, decline to disappearance of snow patches in elevated regions, permafrost degradation, higher rates of evapotranspiration and a decline in summer discharge. To help improve these results for further studies, we suggest completing field campaigns to validate the TC trend analysis, to accurately map permafrost distribution in the studied watershed, as permafrost cannot be directly measured by remote sensing, and, to monitor discharges at the outlet

of the watershed to produce continuous hydrographs.

The method developed and presented here uses open-source data and accessible processing tools, as such, it can be easily reproduced for hydrological studies in other regions of the world. Additionally, the general procedure can also be applied on any type of environmental change assessment study that use unsupervised GeOBIA time series. This study's results will be useful in further research of the environmental monitoring of the GRW, especially in projects focusing on spatial patterns in

water and snow chemistry.





## Appendix A

**Figure A1.** Map of permafrost distribution in Nunavik (northern Quebec, Canada) with the George River watershed boundaries identified by a black line, modified from L'Hérault and Allard (2018). Pink: continuous permafrost, purple: widespread discontinuous permafrost, dark blue: discontinuous permafrost, blue: sporadic permafrost, light blue: isolated to relict permafrost.





**Appendix B**

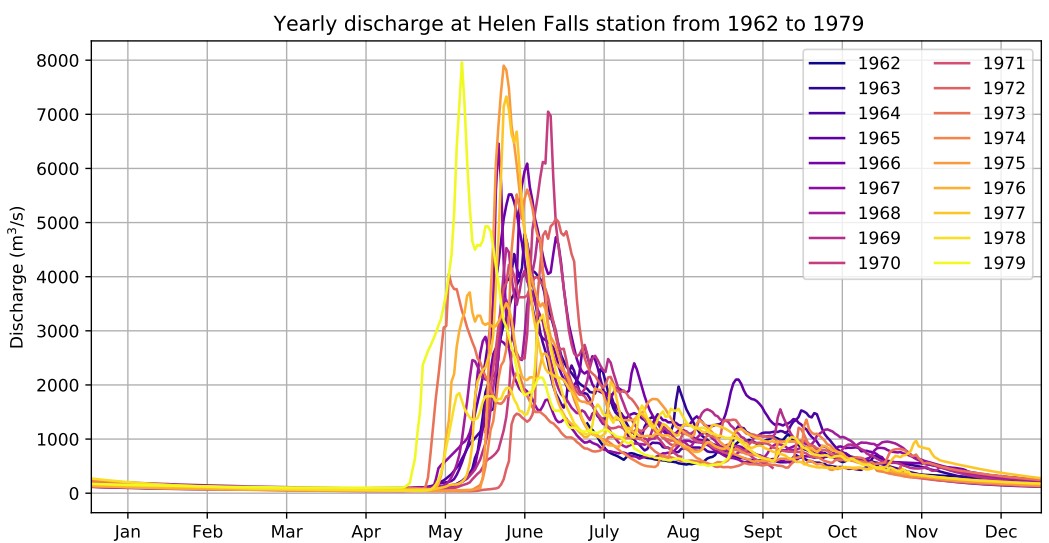

**Figure B1.** Yearly hydrograph of the George River measured at Helen Falls station between 1962 and 1979 (Water level and flow, Environment Canada, Government of Canada).

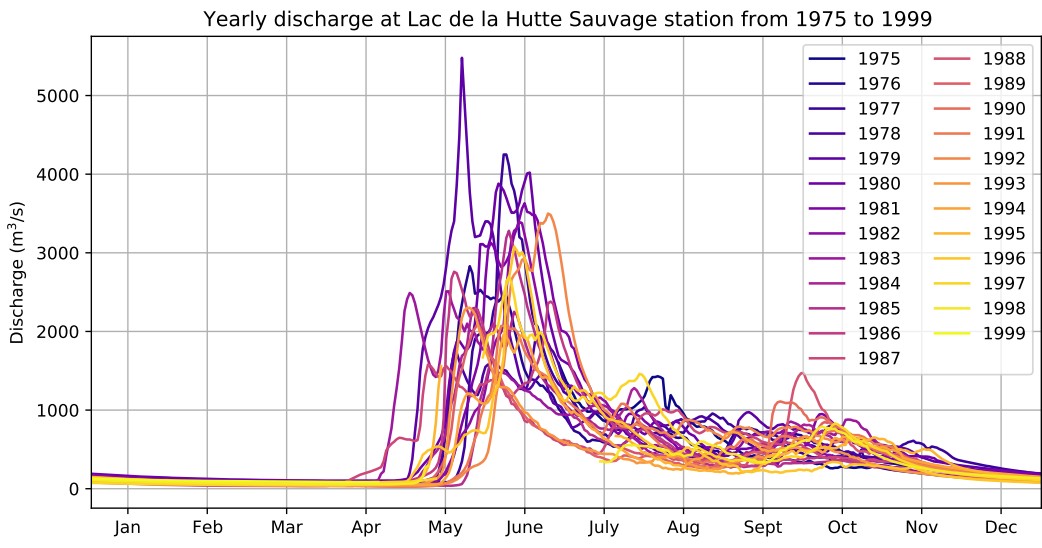

**Figure B2.** Yearly hydrograph of the George River measured at Lac de la Hutte Sauvage station between 1975 and 1999 (Water level and flow, Environment Canada, Government of Canada).





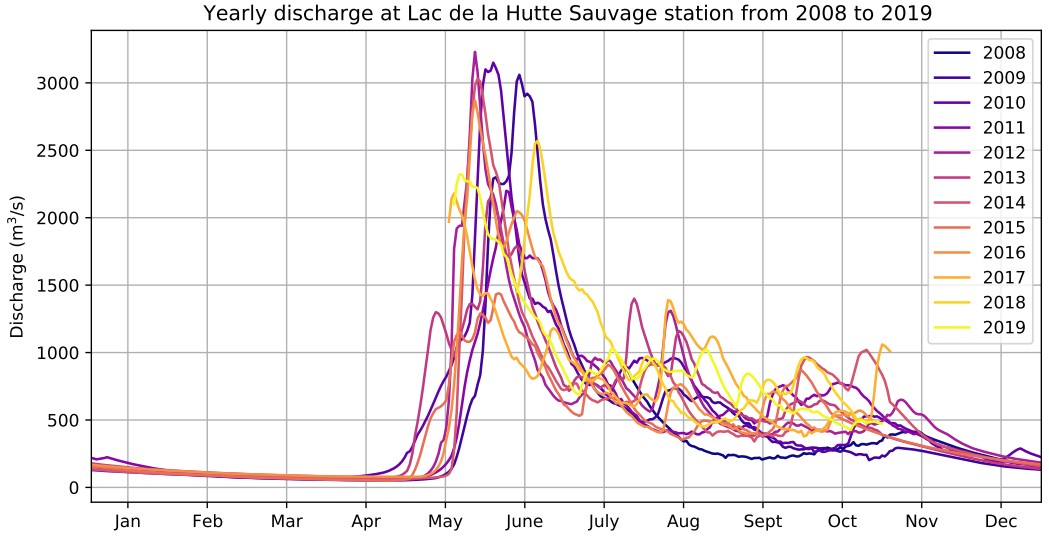

**Figure B3.** Yearly hydrograph of the George River measured at Lac de la Hutte Sauvage station between 2008 and 2019 (Water level and flow, Environment Canada, Government of Canada).

## Appendix C

**Table C1.** Summary of metrics used in the unsupervised Geographic Object-Based Image Analysis. Hydro-geomorphic metrics are categorized into three types: topographic, land cover and geological metrics.

| Metric types | Metrics | Units |
|---|---|---|
| Topographic metrics | Mean elevation | m |
| | Mean slope | ° |
| | Drainage density | $km^{-1}$ |
| | Form factor | - |
| | | |
| Land cover metrics | Mean NDVI | - |
| | NDVI fractional coverage | % |
| | Mean NDMI | - |
| | NDWI fractional coverage | % |
| | | |
| Geological metrics | Thaw-stable deposit cover | % |
| | Thaw-sensitive deposit cover | % |





**Appendix D**

For each normalized index, a mosaic of pixel composite was produced by selecting the median pixel values from the stack of images formed by the collection and clipping the resulting mosaic to the GRW boundaries. Since northern Quebec experiences extensive cloud cover during the growing season, a total of seven mosaics for each index, each spanning a period of five years, were produced to ensure full coverage of the GRW. This whole process was done on the Google Earth Engine (GEE) platform, using JavaScript. Subsequently, we set thresholds in the NDVI and the NDWI, using the mosaics' histogram charts, to clearly

identify vegetated landscape (pixels with NDVI greater than 0.2) and water bodies and snow (pixels with NDWI greater than -0.1). This enabled us to compute the mean NDVI and mean NDMI (while masking the water bodies and snow pixels), the NDVI fractional coverage (percentage of vegetated landscape) and the NDWI fractional coverage (percentage of landscape covered by water bodies or snow). The masking and statistical calculation were processed using a Python script.

**Appendix E**

Membership coefficients are based on the Euclidean distance between the centroids of the objects and the clusters in the 10-dimensional space formed by the metrics. To determine the best belonging cluster of an object, we attributed the cluster associated with the greatest membership coefficient to the object. Because the Fuzzy Partition Coefficient (FPC) can vary greatly depending on the number of clusters one wants to produce, we had to maximize the FPC to find the optimal number of clusters. The FPC maximization was done by trial and error, comparing the classifications' FPCs for 2 to 20 clusters. The

whole clustering analysis was processed using the scikit-fuzzy module (Warner et al., 2019) in Python.

**Appendix F**

The TC Brightness (corresponding to reflectiveness in landscape features), TC Greenness (correlated with vegetation density) and TC Wetness (correlated to water content in water bodies, soil and vegetation) indices were computed using the coefficients provided by Crist and Cicone (1984). TC indices non-parametric linear trends were calculated for every pixel using the Theil-

Sen's method (Theil, 1950; Sen, 1968). To avoid high slope values inferred by single outliers within a year and pixels with high temporal resolution (i.e., multiple values within a year), values outside three standard deviations of each pixel time series were masked and yearly medians were computed for all pixels, improving the original LARCH method (Fraser et al., 2014). Only significant slopes at a 95 % level were selected using the rank-based Mann-Kendall test. The complete TC trend analysis was performed on the GEE platform, using Javascript.





**Appendix G**

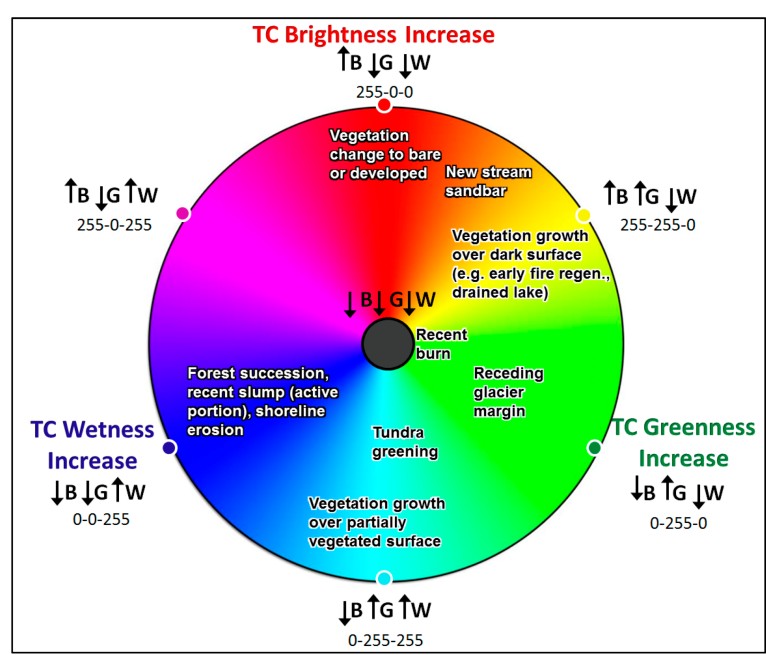

**Figure G1.** Color map legend for Tasseled Cap trend visual interpretation produced by Fraser et al. (2014).

*Code and data availability.* All codes and datasets used in our analysis are publicly available and can be accessed via the references and links we have provided. Please see Table 1 in Sect. 2.2 for relevant links for base-level datasets. Codes and preprocessed datasets are available online at: https://doi.org/10.5281/zenodo.7348972 (Sicaud et al., 2022).

*Author contributions.* ES designed all of the experiments with advice from DF, JPD, and JF. ES conducted all of the experiments and wrote

the manuscript with guidance from DF, JPD and JF. DF and JPD supervised the work and were in charge of the overall direction. Analysis of the results and revision of the manuscript were carried out collectively.

*Competing interests.* The authors declare that they have no conflict of interest.

*Acknowledgements.* We thank ArcticNet (Networks of Centres of Excellence of Canada) for their financial support in this project. As This study is part of the IMALIRIJIIT research program that seeks to integrate scientific approaches and traditional knowledge to advance our



understanding of environment and climate change in Arctic freshwater ecosystems, we would like to thank all the members of the program, especially the Kangiqsualujjuaq community, for their contribution to this study. We also thank OHMi-Nunavik - LabEx DRIIHM, French program Investissements d'Avenir (ANR-11-LABX0010) managed by the French National Research Agency (ANR) for their participation in initiating the program.



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
