# Peer review of "Pairing remote sensing and clustering in landscape hydrology for large-scale changes identification: An application to the subarctic watershed of the George River (Nunavik, Canada)"

_Hydrology and Earth System Sciences, 2023_

## Author Response (AR1)

**Reply to the referee comments**

Reply to referee #1

**C1**: Ln. 101: "*The DEM was first leveled and reconditioned by burning the CanVec lakes and streams shapefiles over it*" It's not clear what you mean. Can you reformulate this sentence and what is the reason why you used the lakes and streams shapefile?

**R1**: The DEM was first leveled and reconditioned by using the CanVec lakes and streams shapefiles. DEM surfaces covered by lakes were flattened and the elevation of pixels covered by streams was negatively accentuated to ensure proper stream convergence, and thus, accurate watersheds delineations. (clarifications applied in the revised manuscript)

**C2**: Ln. 108: There are several ways to define the form factor of a catchment. Which one have you used?

**R2**: We used the form factor defined by Horton (1932) : $F$ = watershed area / (watershed length)$^2$. In our study, we defined the watershed length as the elongation of the polygon forming the watershed delineation. (clarifications applied in the revised manuscript)

**C3**: Ln. 111: It's not clear how the land cover metrics are calculated. The appendix D in itself is very vague and I can not understand how many images you have and how precise is the calculation of the indices. As the process seems quite tedious and long, I suggest to write a section in a supplementary paper.

**R3**: We calculated 4 land cover metrics based on 3 normalized indices : NDVI, NDMI and NDWI. These indices were computed using our collection of Landsat surface reflectance images described in Table 1. The collection is composed of every Landsat-5 TM, Landsat-7 ETM+ and Landsat-8 OLI Collection 2 surface reflectance image with less than 70 % of cloud cover and taken within the growing season (July to August), between 1985 and 2019. Cloudy pixels were masked using the quality assessment (QA) band.

The normalized indices were computed as follow :

NDVI = (NIR – Red) / (NIR + Red)          (Tucker, 1979)

NDMI = (NIR – SWIR1) / (NIR + SWIR1) (Gao, 1996)

NDWI = (Green – NIR) / (Green + NIR)     (McFeeters, 1996)

Since northern Quebec experiences extensive cloud cover during the growing season, a total of 7 mosaics, with a frequency of 5 years, were produced to ensure full coverage of the GRW between 1985 and 2019, for each index. The mosaics were produced by selecting the median pixel values from the stack of images formed by our collection of Landsat images and clipping the resulting mosaic to the GRW boundaries. This whole process was done on the Google Earth Engine (GEE) platform, using JavaScript.

To calculate the 4 land cover metrics based on the normalized indices, we first set thresholds in the NDVI and the NDWI, using the mosaics' histogram charts, to clearly identify vegetated landscape (pixels with NDVI greater than 0.2) and water bodies and snow (pixels with NDWI greater than -0.1). This allowed us to compute for each subwatershed :

1. the mean NDVI while masking the water bodies and snow pixels,
2. the mean NDMI while masking the water bodies and snow pixels,
3. the NDVI fractional coverage defined as the percentage of vegetated landscape, and
4. the NDWI fractional coverage defined as the percentage of landscape covered by water bodies or snow.

The masking and statistical calculation were processed using a Python script. (clarifications applied in the revised manuscript)

**C4**: Ln. 135: What is the frequency of the land cover metrics? From the figure, I guess it is 5 years, but it is not written here. Also, it is not clear what is a seven-classification time series. I guess they are 7 because from 1985 to 2019 there are 35 years divided by 5 years, but this must be explained in the text and not left to the reader's intuition.

**R4**: Yes, the land cover metrics frequency is 5 years. By seven-classification time series we mean a time series composed of 7 classifications. Each classification has also a frequency of 5 years, as they are based on the land cover metrics. The time series spans the time period 1985-2019, for a total of 35 years (7 classifications x 5 years each). (clarifications applied in the revised manuscript)

**C5**: Ln. 140: What is the motivation for using the Tassaled Cup trend analysis instead of other techniques? Please motivate this decision.

**R5**: First, we used the Tasseled Cap trend analysis for its capacity to detect a large variety of land cover changes at once, as it is based on land cover reflectiveness, vegetation abundance and water content trends. Because these trends are respectively presented in the RGB channels of an image, the image resulting from the analysis allows us to identify the nature of the land cover changes simply by looking at the color of the pixels impacted. The brighter the color is, the more important the rate of change is.

Second, the LARCH method using Tasseled Cap trends analysis and presented by Fraser et al. (2014) is specifically designed for land cover changes detection in Arctic and subarctic regions, with a color map (Appendix G) showing the colors associated with the most prevalent types of land cover changes in these regions. This comforted our interpretations on the nature of the land cover changes identified.

Third, the Tasseled Cap trend analysis is well suited for processing on the Google Earth Engine platform, with its Data Catalog containing nearly every Landsat collection available in open access. Performing on the Google Earth Engine platform allowed us to use the same Collection of Landsat surface reflectance images for the clustering analysis and the Tasseled Cap trends analysis, which added coherence in our results and interpretations.

Fourth, the Tasseled Cap trends analysis paired with the processing capability of the Google Earth Engine platform allowed us to detect small scale land cover changes (resolution of 30m) on a vast region (42000 km$^2$) efficiently. As the processing is automatized, the analysis can be applied easily on other vast Arctic and subarctic regions.

We are not aware of any other technique with a versatility and an efficiency matching those of the Tasseled Cap trend analysis in terms of land cover changes detection in Arctic and subarctic regions. (justification applied partly in the revised manuscript)

**Reply to referee #2**

**C1**: Landscape changes are well studied in this manuscript, but hydrological changes seem not well touched. I saw Figure B1-B3 have shown the changes of observed discharge, but the hydrologic data and its connection with landscape changes are not well explored. I suggest to do more analysis to deepen our understanding on landscape hydrological processes.

**R1**: As justified in the Introduction Section 1 (Ln. 70), we decided to use a landscape hydrology approach based only on remotely sensed data because the available hydrological and meteorological data on the George River watershed (GRW) are insufficient for our study, which focus on the period 1985-2019 for the entire GRW:

- The hydrometric station at Helen Falls, based downstream of the George River, which would be most representative of the entire GRW discharge, only acquired data between 1962 and 1979 (before our study period).
- The hydrometric station at Lac de la Hutte Sauvage, based halfway through the George River course, which would be representative of the discharge of the southern half of the GRW only, acquired data during the study period but present an 8-year gap between 2000 and 2007 inclusively (not representative of the entire GRW + large data gap in the middle of our study period).
  Note: There is an error in the manuscript, the first working period of this station is 1975-1999 and not 1975-1996. It has been corrected in the revised version of the manuscript.

Furthermore, the northern half of the GRW is the region that witnessed the most landscape changes during our study period. Thus, we can't explore in detail the connection between landscape changes, where they are the most important, and hydrologic data because of the lack of hydrologic data at Helen Falls station mentioned above.

In other words, the idea behind this paper is not to deepen our understanding on landscape hydrological process, but to present an accessible methodology to detect land cover changes that are known to impact the hydrological regime of remote and ungauged northern watersheds. This is why these potential impacts on a watershed's regime are discussed based on the literature only. (justifications were already presented in the original manuscript, clarification added in the revised manuscript)

**C2**: The phenomenon of greening vegetation, indicated by increasing NDVI, and the decreasing discharge, can be well explained by the increase of root zone storage capacity (SR) (Gao et al., 2014, 10.1002/2014GL061668). As the key element partitioning precipitation into runoff and evaporation, the increase of SR with climate change will increase ecosystem's buffer to drought which sustained aboveground biomass increase, and simultaneously reduced runoff. In another words, larger SR increased the proportion of "green" water, and reduced the "blue" water generation. Another mechanism could be permafrost degradation, which increased subsurface storage capacity, and reduced runoff (Gao et al., 2022, https://doi.org/10.5194/hess-26-4187-2022).

**R2**: We will complete our explanations on the impact of greening vegetation on the discharge in the Discussion Section 4.3 by adding the notions and the reference provided in your comment. The part on permafrost degradation, though, has already been covered in this section. Nevertheless, we will cite the reference provided in your comment to enhance the quality of the discussion. (notions and citations added in the revised manuscript)

**C3**: The terminology of ET. I noticed that the authors cited the paper from Savenije 2004 (https://doi.org/10.1002/hyp.5563), who advocated to delete the term of evapotranspiration. But in the main text and Figure 1, evapotranspiration is still widely used. I would suggest to use term "evaporation" to represent the total evaporation from interception, transpiration, soil evaporation and open-water evaporation.

**R3**: We cited the paper from Savenije (2004) mostly for its notions on the importance of interception in the whole evaporation process. Since we widely discuss the total evaporation in our paper, we agree that it would make sense to follow the ideas and arguments presented by Savenije (2004) and replace the word "evapotranspiration" by "evaporation". (change applied in the revised manuscript)

**C4**: Normally, we don't use sentence as the title. Also don't use period in title.

**R4**: Alternative title (change applied in the revised manuscript):

Pairing remote sensing and clustering in landscape hydrology for large-scale changes identification: An application to the subarctic watershed of the George River (Nunavik, Canada)

Below is a non-exhaustive list of editor's choice, journal highlight, and most cited papers in some high impact factor journals using this formulation for titles. We therefore believe that our new formulation of the title is valid and correct.

Hydrology and Earth System Sciences:

- Quantifying new water fractions and transit time distributions using ensemble hydrograph separation: theory and benchmark tests (Kirchner, 2019)
- Diagnosing hydrological limitations of a land surface model: application of JULES to a deep-groundwater chalk basin (Vine et al., 2016)

- Development of a large-sample watershed-scale hydrometeorological data set for contiguous USA: data set characteristics and assessment of regional variability in hydrologic model performance (Newman et al., 2015)

Journal of Hydrology:
- Real-time probabilistic forecasting of river water quality under data missing situation: Deep learning plus post-processing techniques (Zhou, 2020)
- A review of remote sensing applications for water security: Quantity, quality, and extremes (Chawla et al., 2020)
- Inter-comparison of ERA-5, ERA-interim and GPCP rainfall over the last 40 years: Process-based analysis of systematic and random differences (Nogueira, 2020)

Water Resources Research:
- Landfalling Droughts: Global Tracking of Moisture Deficits From the Oceans Onto Land (Herrera-Estrada and Diffenbaugh, 2020)
- Increasing the Physical Representation of Forest-Snow Processes in Coarse-Resolution Models: Lessons Learned From Upscaling Hyper-Resolution Simulations (Mazzotti et al., 2021)

- Numerical Modeling of Instream Wood Transport, Deposition, and Accumulation in Braided Morphologies Under Unsteady Conditions: Sensitivity and High-Resolution Quantitative Model Validation (Ruiz-Villanueva, 2020)